# Effect of Temperature on Corrosion of L245 Steel Under CO_2_-SRB Corrosion System

**DOI:** 10.3390/microorganisms13030500

**Published:** 2025-02-24

**Authors:** Ming Sun, Xinhua Wang, Wei Cui, Chuntao Shi

**Affiliations:** 1College of Mechanical & Energy Engineering, Beijing University of Technology, Beijing 100124, China; sm8014708@163.com; 2China Special Equipment Inspection & Research Institute, Beijing 100129, China; 13811915141@163.com (W.C.); shicht2024@126.com (C.S.); 3Technology Innovation Center of Oil and Gas Pipeline and Storage Equipment Safety, State Administration for Market Regulation, Beijing 100029, China

**Keywords:** oil and gas gathering pipelines, CO_2_-SRB corrosion system, effect of temperature, L245 pipeline steel

## Abstract

Microorganisms are often observed in the produced medium during the oil and gas extraction process. Corrosion caused by CO_2_ and microorganisms is found on the inner wall of the metal gathering pipelines during the production process. In order to explore the corrosion characteristics of L245 materials under the combined action of sulfate-reducing bacteria (SRB) and CO_2_, a CO_2_-SRB corrosion system was established in this paper. Experimental research on corrosion rate, surface morphology, and corrosion products analysis was conducted. The effect of temperature on the corrosion of SRB while CO_2_ is saturated and the partial pressure is 0.06 MPa was investigated. It was observed that the corrosion is more serious in the CO_2_-SRB corrosion system than that in the single CO_2_ corrosion system. At 40 °C, the corrosion caused by CO_2_ is 0.0597 mm/a, and the corrosion caused by SRB is 0.0766 mm/a. So, more attention should be paid to the corrosion status of gathering pipelines with microorganisms. Further, the activity of SRB is stronger when the temperature of the medium is 40 °C, and corrosion on L245 samples is more obvious under the experimental conditions in this article. In order to reduce the corrosion damage of metal pipelines with microorganisms, the temperature should be well controlled to reduce the activity of SRB during the production process.

## 1. Introduction

The natural gas and associated gas found in the majority of China’s gas fields exhibit distinctive characteristics, including elevated levels of H_2_S and CO_2_, high salinity in formation water, low pH (indicating mild acidity), an increased presence of sulfate-reducing bacteria (SRB), and, to varying extents, iron bacteria [1]. The equipment and pipelines used in production, transportation, and processing are unavoidably exposed to significant corrosion risks, with gathering equipment and pipelines in particular being subjected to considerable challenges. Owing to the multitude of factors contributing to corrosion, along with their complex interactions, the corrosion mechanisms involved are highly intricate. In recent years, there has been growing academic attention on the role of microorganisms in metal corrosion and its underlying mechanisms.

In recent years, with the development of shale gas, the cases of SRB corrosion caused by fracturing technology have increased significantly. The joint action of SRB corrosion with CO_2_ corrosion, Cl^−^ corrosion, and oil–gas erosion can greatly promote the corrosion and perforation of pipelines [2,3,4,5,6,7,8]. As early as 2008, Liu Wei et al. investigated the electrochemical characteristics of X60 corrosion in environments where SRB and CO_2_ coexist [9]. Subsequent studies by researchers have focused on the corrosion behavior of carbon steel, investigating multiple factors such as sand deposition, CO_2_, SRB, and Cl^−^ corrosion [10,11,12,13].

Liao Yubin et al. accounted for the effects of hydrodynamics, temperature, and pressure in developing a corrosion mechanism and prediction model for CO_2_-H2S/SRB [14,15]. As research progresses, the influence of temperature on microbial corrosion has become increasingly significant and cannot be overlooked [16,17,18]. Within the field of oil and gas extraction, CO_2_ corrosion and microbe-induced corrosion are among the most prevalent forms of pipeline degradation. According to the GB17820-2018 [19], the CO_2_ molar fraction in Class II gas must be kept at ≤3%. When the operating pressure reaches 1.3 MPa, the partial pressure of CO_2_ increases to 0.04 MPa, leading to substantial corrosiveness in moist environments. Within relatively anaerobic conditions inside equipment or pipelines, SRB corrosion becomes especially pronounced. In practical operations, CO_2_ and SRB corrosion typically coexist, and their combined mechanisms and interactions are highly complex, making this a prominent and challenging research focus within the industry. Undertaking relevant research is therefore of critical importance.

At present, there are studies on the effects of temperature on bacterial corrosion and CO_2_ partial pressure on corrosion, but there are relatively few studies that comprehensively consider temperature and CO_2_ partial pressure. To investigate the contribution of bacteria and CO_2_ in the joint action of the corrosion process, the following research is conducted.

## 2. CO_2_-SRB Corrosion System Simulation Experiment

The experimental process is shown in the following Figure 1. The first step is bacterial identification and purification. The second step is sample and solution preparation. The third step is sterilization treatment. The fourth step is the corrosion simulation experiment. Then the samples were used to calculate the corrosion rate, observe microscopic morphology, or conduct an electrochemical test.

### 2.1. Water Sample Testing and Strain Identification

In order to carry out a corrosion simulation experiment on the CO_2_-SRB corrosion system, production water was sourced from the gathering system of a gas production well station. The chemical composition of the production water was analyzed using spectrophotometric and titration methods, with the results detailed in Table 1. Bacteria were inoculated into the growth medium for large-scale cultivation, with the medium composition presented in Table 2. Inoculation and cultivation procedures were performed in strict accordance with GB/T 14643.5-2009 [20]. Sulfate-reducing bacteria were isolated using the growth medium, and the strains were identified through 16s rDNA sequencing technology.

The bacterial identification results are illustrated in Figure 2, with Desulfovibrio identified as the predominant species. The bacteria have arc-shaped or rod-shaped cells that appear single or in pairs. Simple organic compounds such as lactate can be used as electron donors and carbon sources for this kind of bacteria, and the bacteria belong to anaerobic bacteria with respiratory metabolism and fermentation metabolism functions [21,22]. As described in the eighth edition of Bergey’s Manual of Systematic Bacteriology, Desulfovibrio is motile via polar flagella, non-spore-forming, Gram-negative, and its optimal growth temperature is influenced by both its origin and history, typically ranging from 25 °C to 30 °C, with an upper limit of 44 °C. Consequently, it is classified as a mesophilic bacterium. The cells are vibroid or rod-shaped, occurring singly or in pairs, and are capable of utilizing simple organic compounds, such as lactate, as electron donors and carbon sources. It is an anaerobic bacterium capable of both respiratory and fermentative metabolism.

Upon completion of the corrosion test, the concentration of sulfate-reducing bacteria (SRB) in the solution was quantified using the most probable number (MPN) method.

### 2.2. Corrosion Weight Loss Test

Corrosion testing was carried out using L245 steel, with its chemical composition detailed in Table 3 below. The test specimens were prepared in three distinct sizes. In accordance with GB/T 19291-2003 [23], the dimensions of the weight loss samples were 60 × 20 × 3 mm, intended for weight loss analysis, with three parallel specimens per group. The morphology observation specimens, measuring 20 × 20 × 3 mm, were utilized for microstructural analysis, while the electrochemical impedance specimens, measuring 10 × 10 × 3 mm, were designated for electrochemical impedance spectroscopy. The specimen surfaces were pretreated using a grinding machine, followed by polishing with #800 grit sandpaper, degreasing with acetone, and subsequent drying under cold air. Finally, the specimens were sterilized under ultraviolet light for 30 min within a laminar flow hood.

Prior to initiating the corrosion experiment, the test containers, PTFE clamps, and deionized water were sterilized in an autoclave at 121 °C for 15 min, while the reagents were sterilized using ultraviolet light in a laminar flow hood for 30 min. The test solutions were formulated by mixing production water with culture medium solutions in varying proportions, specifically at 0%, 10%, 20%, 50%, and 100% concentrations. The prepared solutions underwent deoxygenation by nitrogen gas purging (99.99% purity) for 24 h. The experimental duration was set to 10 days, with a controlled temperature of 40 °C. For each test group, 5 mL of inoculated culture medium was introduced prior to commencement. Some experimental equipment is shown in Figure 3 below.

### 2.3. Characterization of Material Morphology

Upon conclusion of the experiment, the corrosion weight loss specimens were extracted, rinsed with alcohol, and air-dried using cold air. The corrosion products were removed with a pickling solution composed of 500 mL hydrochloric acid, 500 mL deionized water, and 3.5 g hexamethylenetetramine. The specimens were subsequently weighed to determine the corrosion rate, which was calculated based on Equation (1):(1)$$Corrosion\;rate=\frac{K\times W}{A\times T\times D}$$
where *K*—constant, 8.76 × 10^4^;

*T*—corrosion test time, h; 

*A*—exposed area, cm^2^;

*W*—mass loss, g; 

*D*—specimen steel density, g/cm^3^.

The specimens for microstructural observation were extracted and immersed in a 5% glutaraldehyde solution for 8 h to fix surface microorganisms. Subsequently, the specimens underwent stepwise dehydration using ethanol solutions with volume fractions of 20%, 40%, 60%, 80%, and 100%, each for 15 min. After dehydration, the specimens were air-dried and preserved in a desiccator under a nitrogen atmosphere. Microstructural features were examined using a scanning electron microscope (SEM) and analyzed through energy-dispersive X-ray spectroscopy (EDS). Acid-cleaned specimens, with surface films removed, were also observed via SEM to analyze pitting morphology and distribution characteristics. Additionally, 3D pitting dimensions were measured using a three-dimensional stereoscopic microscopy system.

### 2.4. Electrochemical Testing

A three-electrode system was employed to assess the open circuit potential (OCP) and electrochemical impedance spectroscopy (EIS) in order to evaluate variations in the electrochemical state of the specimen surface. The reference electrode used was a saturated calomel electrode (SCE), the auxiliary electrode was a platinum electrode, and the working electrode comprised the L245 steel specimen.

The open circuit potential (OCP) serves as a fundamental parameter in corrosion system studies, capable of characterizing the corrosion state of the metal. By measuring the temporal variation in OCP during the corrosion process under conditions where sulfate-reducing bacteria (SRB) and CO_2_ coexist, a preliminary evaluation of the corrosion progression can be conducted. Nonetheless, the insights provided by OCP data regarding corrosion are limited, yielding predominantly qualitative results; thus, it is imperative to integrate these findings with other electrochemical techniques, such as EIS and polarization curve data, to elucidate the underlying mechanisms driving OCP variations.

The fundamental principle underlying electrochemical impedance spectroscopy (EIS) measurements involves applying a small amplitude voltage (or current) signal to the system while maintaining a DC steady state, subsequently determining the system characteristics through the detection and analysis of the response signals. Throughout the EIS measurement process, a small amplitude electrical signal perturbs the system with minimal influence on its state, rendering it suitable for investigating the effects of microbial processes—such as the adhesion, proliferation, and biofilm formation of sulfate-reducing bacteria (SRB)—on corrosion electrochemical phenomena. The testing parameters included a sine wave excitation signal amplitude of 20 mV, a scanning frequency range of 0.01 Hz to 10⁵ Hz, and the test temperature was aligned with the experimental temperature.

## 3. Corrosion Behavior of L245 Steel in the CO_2_-SRB Corrosion System

Firstly, design a set of experiments to determine the influence of SRB bacteria on the corrosion behavior of the CO_2_ corrosion system as a blank control. The specific experimental conditions are shown in Table 4.

The solution state, odor, and sample condition after the experiment are shown in Table 5.

The surface morphology of the sample after the experiment is shown in Figure 4. The surface of the sterile test group is relatively bright, and there are no obvious corrosion products; the surface of the bacterial test group is covered with a layer of black product. After acid washing and weighing the weightlessness sample, the corrosion rate was calculated as shown in Figure 5. The corrosion rate of the bacteria-containing test group was about twice that of the sterile group, indicating that SRB bacteria played a significant role in the corrosion process.

Scanning electron microscopy observation and EDS energy spectrum analysis were performed on the sample before acid washing, as shown in Figure 6. The surface of the sterile sample still retained obvious scratches, without obvious corrosion products or biofilms. The EDS energy spectrum showed that the main component was Fe, with a small amount of O, indicating that the surface was mainly composed of an iron matrix; the surface of the bacterial sample was covered with clustered products, and the EDS energy spectrum analysis is shown in Table 6. The ESD spectrum shows that the surface components of the bacteria-containing sample include C, O, S, and Fe. Among them, S is important evidence of SRB bacteria participating in corrosion, and the high content of C and O indicates the presence of biological components. Therefore, the surface products are mainly SRB biofilm and corrosion product film formed after SRBs’ participation in corrosion.

By comparing the surface morphology, corrosion rate, scanning electron microscopy, and EDS analysis results of sterile and SRB-containing bacteria under the same experimental conditions, it was found that SRB bacteria have a significant promoting effect on the CO_2_ corrosion system.

## 4. Effect of Temperature on the Corrosion Behavior of L245 Steel

This investigation into the influence of varying temperatures on corrosion mechanisms was conducted under the specific experimental conditions detailed in Table 7 below.

Upon completion of the experiment, the characteristics of the solution, including its state, odor, and the condition of the specimens, are illustrated in Table 8. The surface morphology of the specimens following the experiment is shown in the corresponding figures, where the surfaces of the 20 °C and 60 °C experimental groups exhibit a relatively bright appearance with minimal corrosion products present; in contrast, the surface of the 40 °C experimental group is obscured by a layer of black deposits, as shown in Figure 7 and Figure 8.

Figure 9 shows the content of planktonic bacteria at different temperatures. The bacterial populations were high at 20 °C and 40 °C, while the bacterial population diminished to nearly zero at 60 °C, illustrating the suppressive impact of elevated temperature conditions on bacterial growth. In contrast, the effect of CO_2_ partial pressure on the abundance of planktonic bacteria remains comparatively negligible.

Figure 10 shows the corrosion rate at different temperatures. With an increase in CO_2_ partial pressure to 0.06 MPa, there is a notable escalation in corrosion rates, particularly evident at 60 °C. Under saturated CO_2_ conditions, corrosion is predominantly driven by the activity of SRB bacteria, with peak corrosion rates observed near the optimal growth temperature of 40 °C, as 0.0766 mm/a, while corrosion rates decline at both 20 °C and 60 °C, which is 0.0136 mm/a and 0.0488 mm/a; at a CO_2_ partial pressure of 0.06 MPa, CO_2_ corrosion becomes the dominant mechanism at 60 °C, as 0.7335 mm/a. Solely considering the influence of CO_2_ on bacterial growth, the trend in corrosion rates should correspond to the patterns observed in bacterial abundance, suggesting that the majority of the increased corrosion rates can be attributed to CO_2_-induced corrosion.

At a temperature of 20 °C, bacterial abundance decreases by 80%, resulting in a fivefold increase in the corrosion rate, with the difference attributable to CO_2_ corrosion; at 40 °C, bacterial abundance remains relatively stable, and the corrosion rate is elevated by a factor of 1.78, with the difference also attributed to CO_2_ corrosion; at 60 °C, bacterial abundance is reduced to 60%, while the corrosion rate escalates to fifteen times higher, with the difference indicating CO_2_ corrosion. By employing this analytical framework, the corrosion rates resulting from CO_2_ corrosion are distinctly marked in red, as depicted in the subsequent Figure 11. The figure demonstrates that the corrosion rates at both 20 °C and 40 °C are largely comparable, measured at 0.0572 mm/a and 0.0597 mm/a, respectively, with 40 °C being marginally higher; conversely, at 60 °C, the corrosion rate surges to 0.7042 mm/a, considerably exceeding those observed at other temperatures. Under a CO_2_ partial pressure of 0.06 MPa, 60 °C emerges as a pivotal temperature range for CO_2_ corrosion, accounting for the significant increase in corrosion rates.

Prior to acid washing, scanning electron microscopy (SEM) and energy-dispersive spectroscopy (EDS) analyses were performed on the samples, as depicted in Figure 12, Figure 13, Figure 14 and Figure 15, alongside Table 9 and Table 10. The surface of the sample under the experimental condition of 20 °C exhibited pronounced scratches, with no visible corrosion products or biofilms present; EDS analysis identified the principal components as Fe and C, suggesting that the surface predominantly consists of an iron matrix. The surface of the sample under the experimental condition of 40 °C was covered by clustered products, and EDS analysis revealed the presence of C, O, S, and Fe, with S acting as crucial evidence of SRB bacterial involvement in the corrosion process. The elevated concentrations of C and O indicate the presence of biological constituents, thereby suggesting that the surface products primarily comprise SRB biofilms and corrosion products generated through SRB activity. At 60 °C, the surface displayed a significant product film, with EDS analysis revealing that the primary constituents were Fe and C, likely corresponding to corrosion products formed due to CO_2_ involvement.

Under a CO_2_ partial pressure of 0.06 MPa at 20 °C, the surface biofilm appeared uneven and poorly compacted, exhibiting visible scratches and lacking detectable S in the EDS analysis; at 40 °C, the surface demonstrated biofilm characteristics, with localized clustered corrosion products and a general presence of S, peaking at 7.87%; at 60 °C, a dense FeCO_3_ product film developed, with no S detected on the surface.

Scanning electron microscopy (SEM) observations were conducted on the samples post-acid washing, as illustrated in Figure 16 and Figure 17. Each group of samples exhibited localized corrosion; specifically, the experiment at 20 °C displayed shallow corrosion pits on the surface. In contrast, the experiment at 40 °C revealed a significant number of localized corrosion pits, indicating that the involvement of SRB bacteria not only greatly increased overall corrosion but also resulted in severe localized corrosion. The surface of the experiment at 60 °C showed larger corrosion pits, albeit with reduced depth. Under a CO_2_ partial pressure of 0.06 MPa, the localized corrosion exhibited greater depth and area, characterized by large openings, and closely resembled the corrosion morphology observed under saturated CO_2_ conditions at 60 °C. This observation suggests that the localized corrosion is primarily attributed to CO_2_ corrosion, distinctly differing from the localized corrosion morphology observed under saturated CO_2_ conditions at 40 °C.

The results of the open circuit potential and polarization curve tests are illustrated in Figure 18. It is evident that under the optimal growth conditions for bacteria at 40 °C, the corrosion current density reaches its maximum. Conversely, as the temperature increases to 60 °C, the polarization curve shifts to the left, indicating a reduction in corrosion current density. At 20 °C, the corrosion current density is at its lowest, consistent with the overall corrosion results.

To further investigate the characteristics of the surface film layer on the samples, electrochemical testing was conducted using electrochemical impedance spectroscopy (EIS), and key electrochemical parameters were analyzed by fitting the EIS curves. The results of the impedance fitting are illustrated in Figure 19, with the fitting parameters detailed in Table 11. The circuit model employed for the fitting aligns with the double time constant circuit model depicted in Figure 20, where the critical parameter *R_bc_ + R_ct_* reflects the susceptibility of the electrochemical circuit to corrosion.

The fitting parameters *R_bc_ + R_ct_* from the electrochemical impedance analysis and the corrosion rate data are plotted together in Figure 21 below, revealing an inverse correlation between the corrosion rate and *R_bc_ + R_ct_*.

At 20 °C, the overall corrosion rate is the lowest, with planktonic bacterial content comparable to that at 40 °C, alongside some localized corrosion; at 40 °C, the overall corrosion rate is the highest, with a complete film formation and the most pronounced pitting, indicating the most significant role of bacteria in the corrosion process, similar to Fan’s research findings [24]; at 60 °C, bacterial growth is inhibited, resulting in the lowest planktonic bacterial content. However, due to the effects of saturated CO_2_, the corrosion rate is higher than that at 20 °C, accompanied by localized corrosion. The research results indicate that specific types of bacteria have suitable temperature ranges. If the operating temperature is within the identified suitable temperature range for bacteria, monitoring of bacterial corrosion should be strengthened, and consideration should be given to increasing the injection of fungicides.

## 5. Conclusions

Compared with the CO_2_ corrosion system alone, the CO_2_-SRB corrosion system significantly enhances bacterial clustering and corrosiveness, and the corrosion rate can be increased by about one time.Through weight loss analysis, electrochemical testing, and surface morphology observations, the corrosion severity trends align consistently. When the corrosion rate is high, distinct corrosion pits are observed on the surface of the experimental specimens.Under the experimental conditions and the produced water used in this study, microbial corrosion intensifies with increasing temperature, reaching peak microbial activity at 40 °C.Based on the experimental findings, it is recommended to control the medium temperature in the CO_2_-SRB corrosion system to mitigate corrosion. If the operating temperature is within the suitable temperature range for identifying bacteria, monitoring of bacterial corrosion should be strengthened, and consideration should be given to increasing the injection of fungicides.This study only investigated the situation of Desulfovibrio and did not consider the universality of other common SBR bacteria. Further research should supplement relevant experimental results.

## Figures and Tables

**Figure 1 microorganisms-13-00500-f001:**
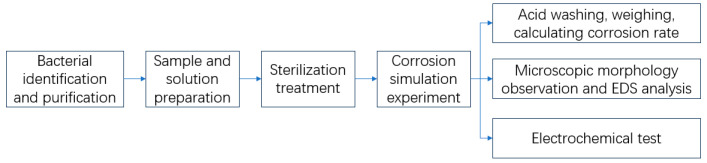
The experimental process.

**Figure 2 microorganisms-13-00500-f002:**

Identification results of bacteria in produced water of gas well station.

**Figure 3 microorganisms-13-00500-f003:**
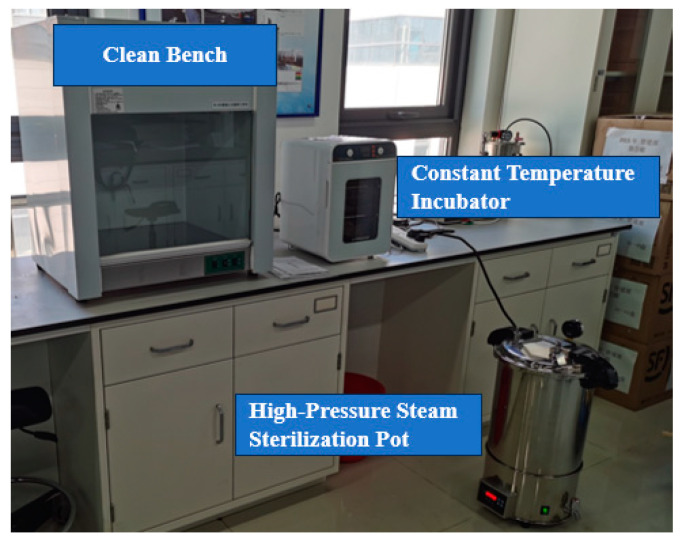
Partial experimental equipment used.

**Figure 4 microorganisms-13-00500-f004:**
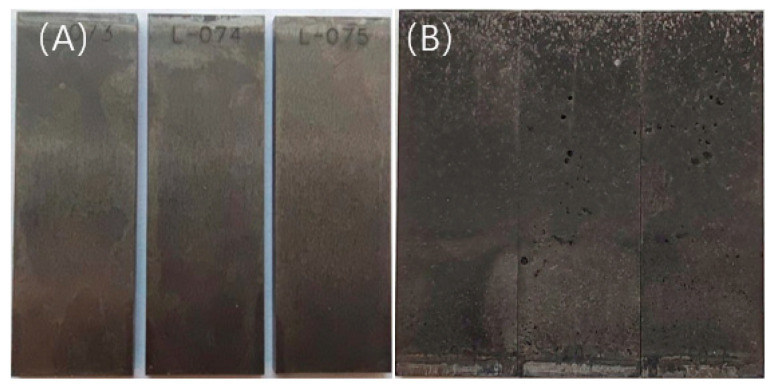
Macro-morphology observation under different bacterial inoculation conditions: (**A**) sterility; (**B**) containing bacteria.

**Figure 5 microorganisms-13-00500-f005:**
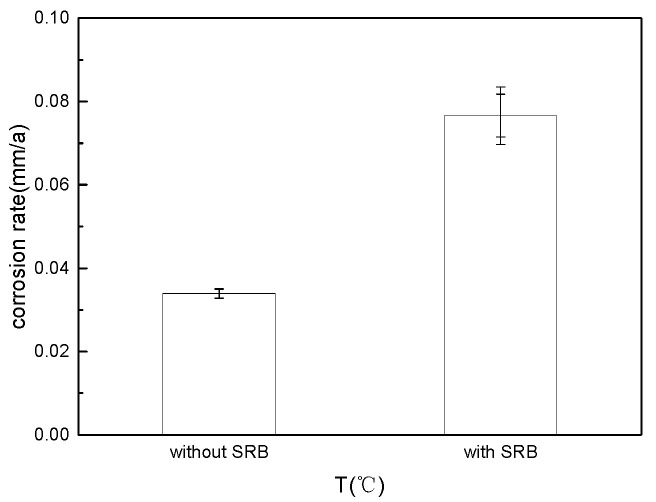
Corrosion rate under different bacterial inoculation conditions.

**Figure 6 microorganisms-13-00500-f006:**
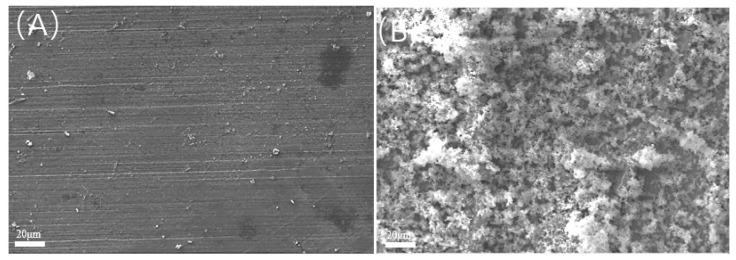
SEM observation of samples before pickling under different bacterial inoculation conditions: (**A**) sterility; (**B**) containing bacteria.

**Figure 7 microorganisms-13-00500-f007:**
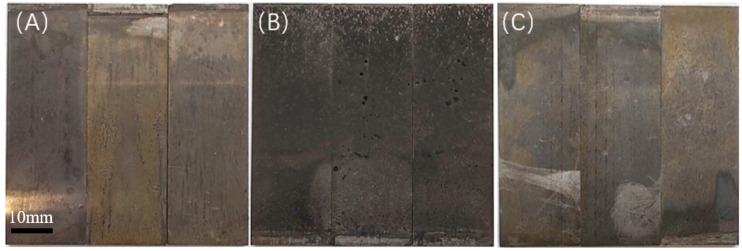
Macroscopic morphology observation under different temperature conditions (saturated CO_2_): (**A**) 20 °C; (**B**) 40 °C; (**C**) 60 °C.

**Figure 8 microorganisms-13-00500-f008:**
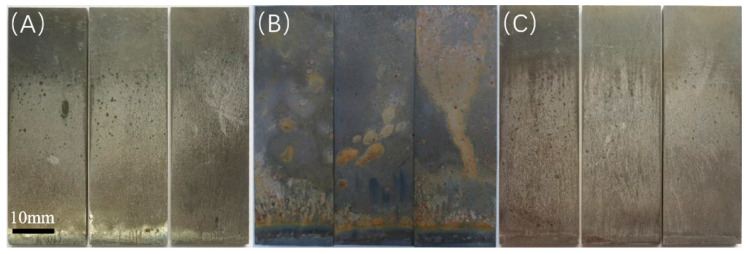
Macroscopic morphology observation under different temperature conditions (partial pressure of CO_2_ 0.06 MPa): (**A**) 20 °C; (**B**) 40 °C; (**C**) 60 °C.

**Figure 9 microorganisms-13-00500-f009:**
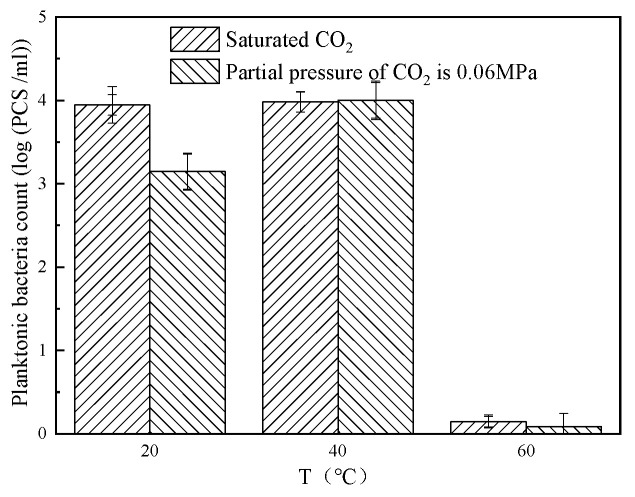
Content of planktonic bacteria at different temperatures.

**Figure 10 microorganisms-13-00500-f010:**
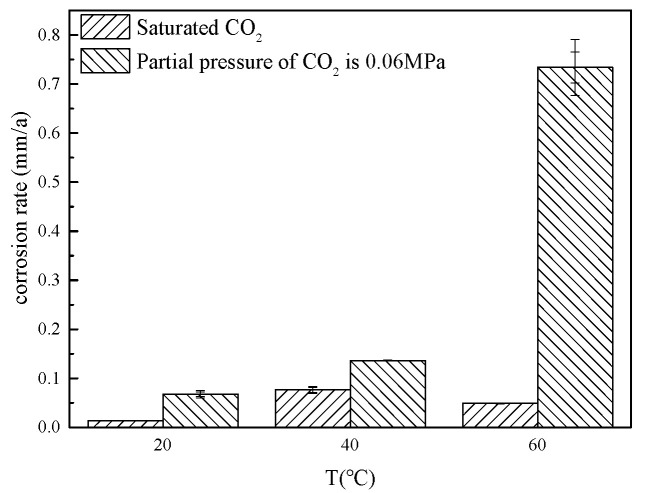
Corrosion rate at different temperatures.

**Figure 11 microorganisms-13-00500-f011:**
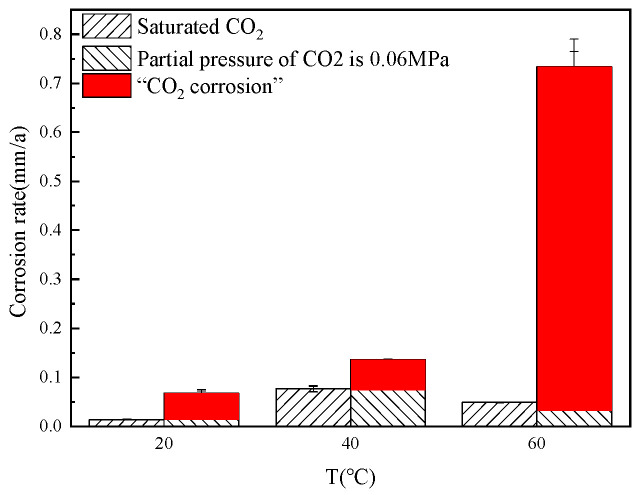
Corrosion rate at different temperatures, the corrosion rates resulting from CO_2_ corrosion are distinctly marked in red.

**Figure 12 microorganisms-13-00500-f012:**
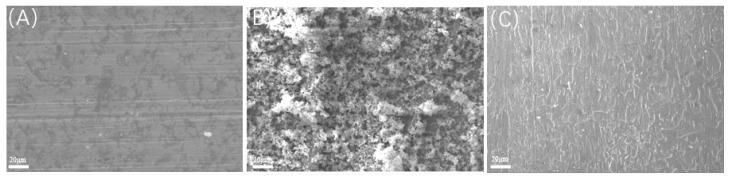
Scanning electron microscopic observation of samples before pickling at different temperatures (saturated CO_2_): (**A**) 20 °C; (**B**) 40 °C; (**C**) 60 °C.

**Figure 13 microorganisms-13-00500-f013:**
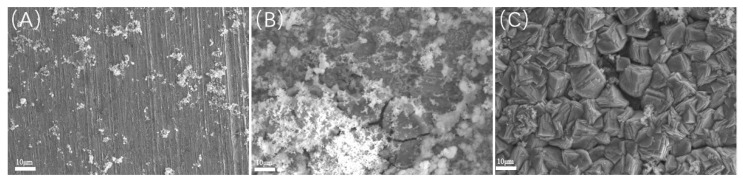
Scanning electron microscopic observation of samples before pickling at different temperatures (partial pressure of CO_2_ 0.06 MPa): (**A**) 20 °C; (**B**) 40 °C; (**C**) 60 °C.

**Figure 14 microorganisms-13-00500-f014:**
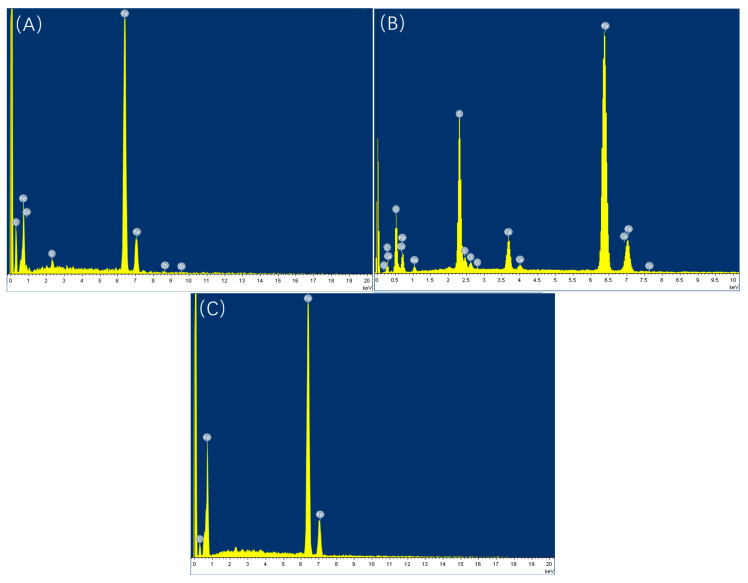
EDS energy spectrum analysis of samples before pickling at different temperatures (saturated CO_2_): (**A**) 20 °C; (**B**) 40 °C; (**C**) 60 °C.

**Figure 15 microorganisms-13-00500-f015:**
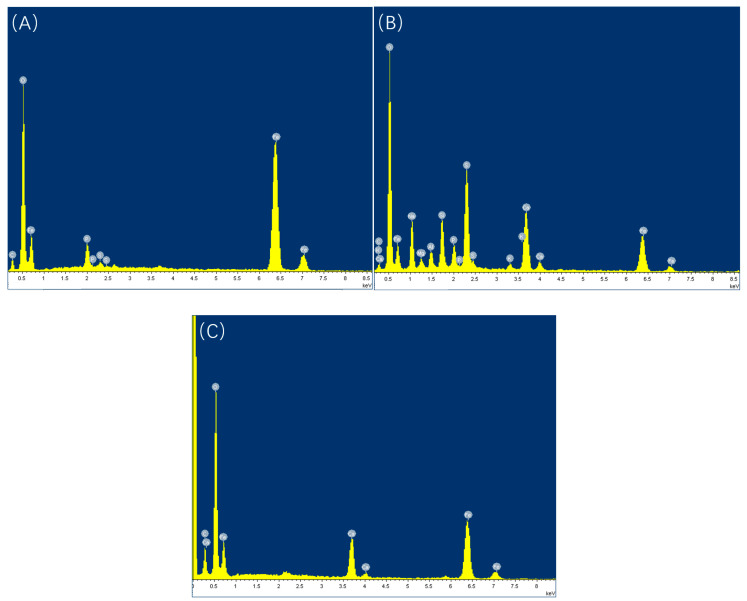
EDS energy spectrum analysis of samples before pickling at different temperatures (partial pressure of CO_2_ is 0.06 MPa): (**A**) 20 °C; (**B**) 40 °C; (**C**) 60 °C.

**Figure 16 microorganisms-13-00500-f016:**
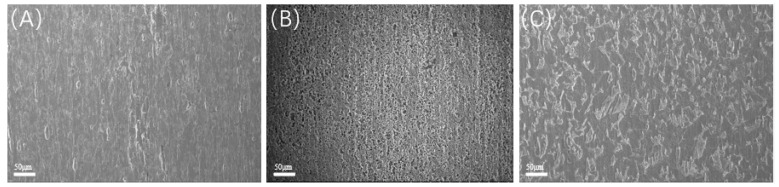
SEM observation of samples after pickling at different temperatures (saturated CO_2_): (**A**) 20 °C; (**B**) 40 °C; (**C**) 60 °C.

**Figure 17 microorganisms-13-00500-f017:**
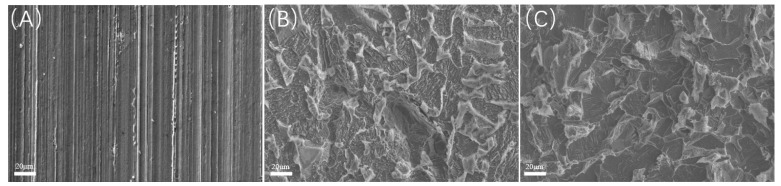
SEM observation of samples after pickling at different temperatures (partial pressure of CO_2_ is 0.06 MPa): (**A**) 20 °C; (**B**) 40 °C; (**C**) 60 °C.

**Figure 18 microorganisms-13-00500-f018:**
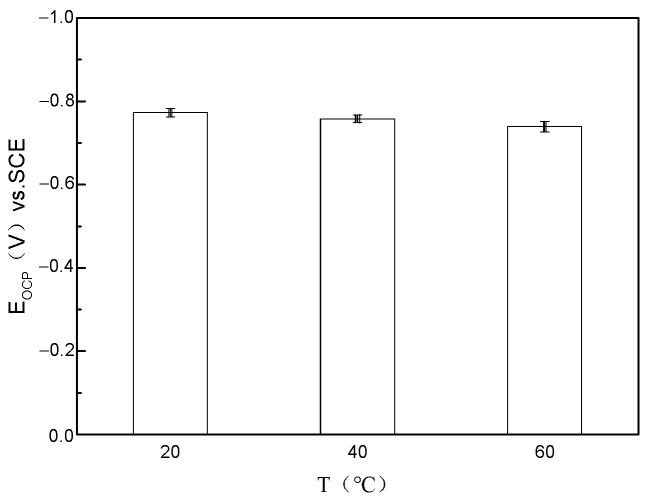
Open circuit potential test under different temperature conditions.

**Figure 19 microorganisms-13-00500-f019:**
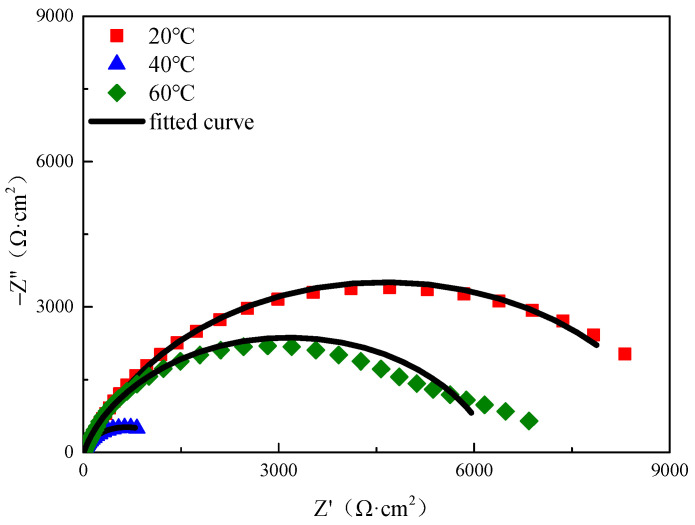
Nyquist diagram of AC impedance under different temperature conditions.

**Figure 20 microorganisms-13-00500-f020:**
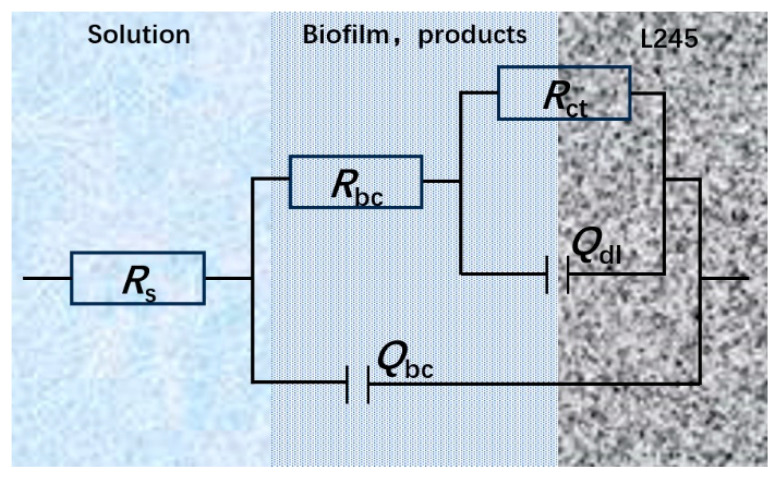
Equivalent circuits for EIS fitting.

**Figure 21 microorganisms-13-00500-f021:**
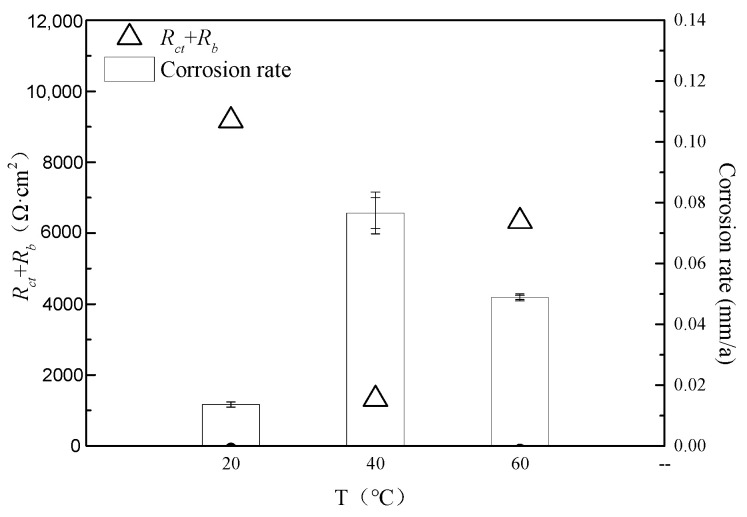
Comparison of AC impedance fitting parameters and corrosion rate under different temperature conditions.

**Table 1 microorganisms-13-00500-t001:** Composition of produced water.

Ingredient	NaCl	CaCl_2_	MgCl_2_·6H_2_O	Na_2_SO_4_	NaHCO_3_
Mass, mg/L	32,654	3301	926	130	225

**Table 2 microorganisms-13-00500-t002:** Composition of culture medium.

Ingredient	K_2_HPO_4_·3H_2_O	NH_4_Cl	Na_2_SO_4_	CaCl_2_	MgSO_4_	C_3_H_5_NaO_3_	C_15_H_31_N3O_13_P_2_	Fe(NH_4_)_2_·(SO_4_)_2_	C_6_H_8_O_6_
Mass, mg/L	500	1000	500	100	2000	3500	1000	300	100

**Table 3 microorganisms-13-00500-t003:** Composition of L245 steel.

Mass, %	C	Si	Mn	P	S	Cr	Ni	Mo	V	Nb	Ti
L245 steel	0.203	0.255	0.397	0.0189	0.0139	0.023	0.021	0.019	0.0015	0.003	0.0025

**Table 4 microorganisms-13-00500-t004:** Experimental parameters under different bacterial inoculation conditions.

Serial Number	Temperature, °C	CO_2_ Pressurization, MPa	Timing, Day	SRB
1	40	saturated	10	no
2	40	saturated	10	yes

**Table 5 microorganisms-13-00500-t005:** Experimental phenomena under different bacterial inoculation conditions.

SRB	Temperature, °C	Odors	Solution	Precipitates
no	40	no rotten egg odor	yellowish	yellowish
yes	40	slightly rotten egg odor	yellowish	yellow

**Table 6 microorganisms-13-00500-t006:** EDS spectrum analysis of samples before pickling under different bacterial inoculation conditions.

SRB	C	O	Na	Si	S	Cl	Ca	Mn	Fe	Co	Ni	Total Amount
no	0	3.53	0	0.31	0	0	0	0.51	95.51	0	0.14	100
yes	9.82	15.35	1.12	0	12.01	0.67	3.05	0	55.5	2.49	0	100

**Table 7 microorganisms-13-00500-t007:** Test parameters at different temperature conditions.

Serial Number	Temperature, °C	CO_2_ Pressurization, MPa	Timing, Day	SRB
1	20	saturated	10	yes
2	40	saturated	10	yes
3	60	saturated	10	yes
4	20	0.06	10	yes
5	40	0.06	10	yes
6	60	0.06	10	yes

**Table 8 microorganisms-13-00500-t008:** Experimental phenomena at different temperature conditions.

Temperature, °C	Odors	Solution	Precipitates
20	slightly rotten egg odor	white	none
40	slightly rotten egg odor	yellowish	yellow
60	no rotten egg odor	white	none
20	slightly rotten egg odor	yellowish	Light yellow precipitate
40	slightly rotten egg odor	yellowish	precipitate of earthy yellow color
60	no rotten egg odor	White to yellow	tan

**Table 9 microorganisms-13-00500-t009:** EDS energy spectrum analysis of samples before pickling at different temperatures (saturated CO_2_).

Temperature, °C	20	40	60
C	29.48	9.82	11.31
O	0	15.35	0
Na	0	1.12	0
S	0.88	12.01	0
Cl	0	0.67	0
Ca	0	3.05	0
Fe	69	55.5	88.69
Co	0	2.49	0
Zn	0.64	0	0
Total amount	100	100	100

**Table 10 microorganisms-13-00500-t010:** EDS energy spectrum analysis of samples before pickling at different temperatures (partial pressure of CO_2_ is 0.06 MPa).

Temperature, °C	20	40	60
C	7.63	5.8	13.76
O	43.5	55.8	58.95
Na	0	6.63	0
Mg	0	0.8	0
Al	0	1.32	0
Si	0	3.56	0
P	3.03	1.96	0
S	0.58	7.87	0
K	0	0.6	0
Ca	0	6.2	5.73
Fe	45.27	9.45	21.56
Total amount	100	100	100

**Table 11 microorganisms-13-00500-t011:** Fitting parameters of AC impedance under different temperature conditions.

Temperature, °C	*R_s_*, Ω·cm^2^	Qbc, Ω^−1^·cm^−2^·S^n^	Rb, Ω·cm^2^	Qdl, Ω^−1^·cm^−2^·S^n^	Rct, Ω·cm^2^	Rbc + Rct, Ω·cm^2^
20	19.13	2.029 × 10^−4^	628.4	1.758 × 10^−4^	8546	9174.4
40	3.224	4.497 × 10^−3^	31.93	1.9 × 10^−3^	1282	1313.93
60	7.655	2.093 × 10^−4^	12.23	1.942 × 10^−5^	6320	6332.23

## Data Availability

The original contributions presented in this study are included in the article. Further inquiries can be directed to the corresponding author.

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
