# Peer review of "Effect of Temperature on Corrosion of L245 Steel Under CO2-SRB Corrosion System"

_microorganisms, 2025, doi:10.3390/microorganisms13030500_

Round 1
Reviewer 1 Report
Comments and Suggestions for Authors
Dear authors, After carefully studying your article, I decided that it can be published only after significant changes to its content. The article is very small and should be expanded by increasing the number of sections. Below I have described in detail my suggestions for improving the article.
1. The abstract should pay more attention to the results of the study in numerical form as well as the practical implementation of the results obtained.
2. In the Introduction section, more attention should be paid not only to Chinese but also to global experience. In addition, the Introduction section is very small and should be significantly expanded. At the end of the section, it is necessary to provide the research gap, the main aim, and the main objectives of the study.
3. It is desirable to show the sequence of experiments in the form of a block diagram. In Section 2, it is also welcomed to present photographs of samples and/or laboratory equipment used.
4. In all figures and in all tables, it is desirable to show the errors in determining the indicators.
5. The work will benefit significantly from the addition of kinetic calculations.
6. Figures 2 and 3 require scales with units of measurement.
7. Why were the parameter values ​​given in Table 4 chosen for the experiments?
8. There is no comparison of the results obtained with those obtained by other authors.
9. The conclusions need to be significantly expanded by adding the purpose of the study, methodology, results of the study in numerical form, limitations, and plans for future research.
10. It is necessary to add references from 2023-2025.
Author Response
We are grateful to the Editor and Reviewers for their support of this manuscript. These comments add to the richness and relevance of this work. We carefully revised every comment from the Editor and Reviewers. Misrepresentations and inaccurate descriptions in the manuscript have been corrected and clarifications added. Modifications and revisions in the manuscript have been highlighted. Detailed responses to the Editor and Reviewers are provided below.
- The abstract should pay more attention to the results of the study in numerical form as well as the practical implementation of the results obtained.
Response 1: It has been revised.
- In the Introduction section, more attention should be paid not only to Chinese but also to global experience. In addition, the Introduction section is very small and should be significantly expanded. At the end of the section, it is necessary to provide the research gap, the main aim, and the main objectives of the study.
Response 2: It has been revised.
- It is desirable to show the sequence of experiments in the form of a block diagram. In Section 2, it is also welcomed to present photographs of samples and/or laboratory equipment used.
Response 3: It has been revised.
- In all figures and in all tables, it is desirable to show the errors in determining the indicators.
Response 4: It has been revised.
- The work will benefit significantly from the addition of kinetic calculations.
Response 5: I’m sorry there’s no kinetic calculations in this study.
- Figures 2 and 3 require scales with units of measurement.
Response 6: It has been revised.
- Why were the parameter values ​​given in Table 4 chosen for the experiments?
Response 7: Refer to newly added references 8 and 9.
- There is no comparison of the results obtained with those obtained by other authors.
Response 8: It has been revised.
- The conclusions need to be significantly expanded by adding the purpose of the study, methodology, results of the study in numerical form, limitations, and plans for future research.
Response 9: It has been revised.
- It is necessary to add references from 2023-2025.
Response 10: It has been revised.

Reviewer 2 Report
Comments and Suggestions for Authors
Dear Authors:
I checked your manuscript closely. The manuscript is well organized, but I would like you to modify and revise your manuscript from the following viewpoints.
#1: Introduction:
The introduction provides a good foundation, but it could be strengthened by including more recent and relevant studies on CO2-SRB corrosion mechanisms. Adding these references will help place your work in a broader scientific context and highlight its importance.
#2: Methods:
The description of microbial cultivation and strain selection could be more detailed. For instance, explaining the isolation and identification process for Desulfovibrio would make it easier for others to reproduce your work.
It would also help to clarify why specific experimental conditions, such as the CO2 partial pressure and temperature range, were chosen. This will provide a clearer justification for your study design.
#3: Results:
While the results are interesting, the contributions of CO2-induced and SRB-induced corrosion could be separated more clearly in your analysis. This would give readers a better understanding of how these factors interact.
Some figures, like Figures 5 and 6, could use more explanation in the text. For example, you could describe how they relate to the trends in corrosion rates and what key insights they reveal.
#4: Discussion:
The discussion is thorough but could benefit from exploring the broader implications of your findings. For instance, how might your observations influence decisions about materials or the development of corrosion inhibitors in the oil and gas industry?
It would also be helpful to explain in more detail why SRB activity peaks at 40°C and how this aligns with findings from other research.
#5: Language and Clarity:
While the manuscript is generally clear, there are some minor grammatical issues and areas where the phrasing could be simplified. I suggest a careful review or professional editing to ensure your ideas come across smoothly.
#6:Conclusion:
The conclusion does a good job summarizing the findings, but it could emphasize their practical importance a bit more. For example, you might discuss how controlling temperature or using specific coatings could mitigate corrosion in real-world applications.
Comments on the Quality of English LanguageThe English quality of this manuscript is adequate for understanding the research. However, there are areas where the language could be polished to improve clarity and presentation. While the authors have expressed their ideas effectively, minor grammatical corrections and rephrasing in complex sections, particularly in the results and discussion, would enhance readability. A professional language editing service could be considered before final publication.
Author Response
We are grateful to the Editor and Reviewers for their support of this manuscript. These comments add to the richness and relevance of this work. We carefully revised every comment from the Editor and Reviewers. Misrepresentations and inaccurate descriptions in the manuscript have been corrected and clarifications added. Modifications and revisions in the manuscript have been highlighted. Detailed responses to the Editor and Reviewers are provided below.
#1: Introduction:
The introduction provides a good foundation, but it could be strengthened by including more recent and relevant studies on CO2-SRB corrosion mechanisms. Adding these references will help place your work in a broader scientific context and highlight its importance.
Response 1: It has been revised.
#2: Methods:
The description of microbial cultivation and strain selection could be more detailed. For instance, explaining the isolation and identification process for Desulfovibrio would make it easier for others to reproduce your work.
It would also help to clarify why specific experimental conditions, such as the CO2 partial pressure and temperature range, were chosen. This will provide a clearer justification for your study design.
Response 2: It has been revised.
#3: Results:
While the results are interesting, the contributions of CO2-induced and SRB-induced corrosion could be separated more clearly in your analysis. This would give readers a better understanding of how these factors interact.
Some figures, like Figures 5 and 6, could use more explanation in the text. For example, you could describe how they relate to the trends in corrosion rates and what key insights they reveal.
Response 3: It has been revised.
#4: Discussion:
The discussion is thorough but could benefit from exploring the broader implications of your findings. For instance, how might your observations influence decisions about materials or the development of corrosion inhibitors in the oil and gas industry?
It would also be helpful to explain in more detail why SRB activity peaks at 40°C and how this aligns with findings from other research.
Response 4: It has been revised.
#5: Language and Clarity:
While the manuscript is generally clear, there are some minor grammatical issues and areas where the phrasing could be simplified. I suggest a careful review or professional editing to ensure your ideas come across smoothly.
Response 5: It has been revised.
#6:Conclusion:
The conclusion does a good job summarizing the findings, but it could emphasize their practical importance a bit more. For example, you might discuss how controlling temperature or using specific coatings could mitigate corrosion in real-world applications.
Response 6: It has been revised.

Reviewer 3 Report
Comments and Suggestions for Authors
INTEREST:
Dear Editor,
The manuscript entitled "Effect of Temperature on Corrosion of L245 Steel under CO2-SRB Corrosion System" by Sun Ming and co-workers presents an interesting study on the combined effects of CO2 and sulfate-reducing bacteria (SRB) on pipeline steel corrosion. This topic is highly relevant for the field of microbiologically influenced corrosion, particularly in the context of the oil and gas industry. The study employs a combination of experimental approaches, including weight loss analysis, SEM, EDS, and electrochemical testing, to investigate corrosion mechanisms under varied temperature conditions.
However, before the manuscript can be considered for publication in Microorganisms Journal from MDPI, several critical aspects of the manuscript need to be improved to meet the journal's quality standards. I recommend a major revision to address the following concerns:
- Clarity and Precision: The abstract and introduction lack clarity in presenting the key findings and their significance. Quantitative results should be emphasized to highlight the study’s contributions.
- Depth of Discussion: The discussion section does not adequately interpret the experimental findings or critically compare them with the existing literature. More in-depth analysis of the temperature-dependent transition from microbial to CO2-driven corrosion is necessary.
- Statistical Robustness: The absence of statistical analysis, such as error bars or standard deviations, weakens the reliability of the findings. Statistical significance should be assessed and reported for all key results.
- Language and Grammar: The manuscript contains several grammatical errors and instances of awkward phrasing, which affect readability. A thorough language revision is recommended.
- Conclusions and Practical Implications: The conclusions should provide actionable insights and recommendations for mitigating CO2-SRB corrosion in practical applications.
Please see my detailed comments report below for further guidance on improving the manuscript.
DETAILED COMMENTS:
Important points for the authors to consider:
1. The abstract does not clearly state the key findings or their significance. For instance, while the effect of temperature is highlighted, there is no quantitative comparison provided. Example: "Corrosion is more serious in CO2-SRB corrosion system than that in single CO2 corrosion system" lacks a specific metric or data point for comparison.
2. The manuscript does not discuss whether the results (e.g., weight loss, EIS measurements) were statistically analyzed. Including error bars, standard deviations, or statistical tests would strengthen the validity of the findings. In electrochemical systems, it is well known that reproducibility of data is always an issue and the authors need to state this (mandatory!!!)
3. The discussion section inadequately addresses why the corrosion rate peaks at 40°C and declines at higher temperatures. The role of bacterial metabolism and biofilm formation is mentioned but not sufficiently linked to the experimental findings. Similarly, the transition from microbial to CO2-driven corrosion at higher temperatures deserves more attention.
4. Figure 6 shows that corrosion at 60°C is primarily driven by CO2. However, the text (page 7) states that bacterial growth is significantly inhibited, which could imply minimal microbial contribution. This inconsistency should be clarified.
5. Figures such as the Nyquist diagrams (Figure 14) lack adequate explanations in the main text. For example, the relationship between fitting parameters and corrosion behavior should be explicitly stated rather than implied. Tables summarizing EDS data (e.g., Table 6) would benefit from a direct comparison to the corrosion rates at corresponding temperatures.
6. The introduction briefly mentions prior studies but does not critically compare them to this work. For example, how does this study advance knowledge compared to Liu et al. (2008) or Eduok et al. (2019)? In addition, the authors could better highlight the novelty of their findings.
7. There are grammatical errors and awkward phrasing throughout the manuscript. For example: (i) "Corrosion caused by CO2 and microorganisms were found on the inner wall..." (page 1): "were" should be "was." (ii) "In order to explore the corrosion characteristics... were investigated" (page 1): Consider rephrasing for conciseness. Improving language clarity would significantly enhance readability.
8. The conclusions reiterate findings without offering actionable insights or broader implications. For example, specific recommendations for mitigating CO2-SRB corrosion in practical applications could be provided.
Comments on the Quality of English LanguagePlease see my detailed comments report for further guidance on improving the manuscript.
Author Response
We are grateful to the Editor and Reviewers for their support of this manuscript. These comments add to the richness and relevance of this work. We carefully revised every comment from the Editor and Reviewers. Misrepresentations and inaccurate descriptions in the manuscript have been corrected and clarifications added. Modifications and revisions in the manuscript have been highlighted. Detailed responses to the Editor and Reviewers are provided below.
- The abstract does not clearly state the key findings or their significance. For instance, while the effect of temperature is highlighted, there is no quantitative comparison provided. Example: "Corrosion is more serious in CO2-SRB corrosion system than that in single CO2 corrosion system" lacks a specific metric or data point for comparison.
Response 1: It has been revised.
- The manuscript does not discuss whether the results (e.g., weight loss, EIS measurements) were statistically analyzed. Including error bars, standard deviations, or statistical tests would strengthen the validity of the findings. In electrochemical systems, it is well known that reproducibility of data is always an issue and the authors need to state this (mandatory!!!)
Response 2: It has been revised.
- The discussion section inadequately addresses why the corrosion rate peaks at 40°C and declines at higher temperatures. The role of bacterial metabolism and biofilm formation is mentioned but not sufficiently linked to the experimental findings. Similarly, the transition from microbial to CO2-driven corrosion at higher temperatures deserves more attention.
Response 3: It has been revised.
- Figure 6 shows that corrosion at 60°C is primarily driven by CO2. However, the text (page 7) states that bacterial growth is significantly inhibited, which could imply minimal microbial contribution. This inconsistency should be clarified.
Response 4: It has been revised.
- Figures such as the Nyquist diagrams (Figure 14) lack adequate explanations in the main text. For example, the relationship between fitting parameters and corrosion behavior should be explicitly stated rather than implied. Tables summarizing EDS data (e.g., Table 6) would benefit from a direct comparison to the corrosion rates at corresponding temperatures.
Response 5: It has been revised.
- The introduction briefly mentions prior studies but does not critically compare them to this work. For example, how does this study advance knowledge compared to Liu et al. (2008) or Eduok et al. (2019)? In addition, the authors could better highlight the novelty of their findings.
Response 6: It has been revised.
- There are grammatical errors and awkward phrasing throughout the manuscript. For example: (i) "Corrosion caused by CO2 and microorganisms were found on the inner wall..." (page 1): "were" should be "was." (ii) "In order to explore the corrosion characteristics... were investigated" (page 1): Consider rephrasing for conciseness. Improving language clarity would significantly enhance readability.
Response 7: It has been revised.
- The conclusions reiterate findings without offering actionable insights or broader implications. For example, specific recommendations for mitigating CO2-SRB corrosion in practical applications could be provided.
Response 8: It has been revised.

Round 2
Reviewer 1 Report
Comments and Suggestions for Authors
This version of the article can be published in the journal.